# Optimization of Asphalt-Mortar-Aging-Resistance-Modifier Dosage Based on Second-Generation Non-Inferior Sorting Genetic Algorithm

**DOI:** 10.3390/ma15103635

**Published:** 2022-05-19

**Authors:** Yang Lv, Shaopeng Wu, Peide Cui, Serji Amirkhanian, Haiqin Xu, Yingxue Zou, Xinkui Yang

**Affiliations:** 1State Key Laboratory of Silicate Materials for Architectures, Wuhan University of Technology, Wuhan 430070, China; lvyang@whut.edu.cn (Y.L.); cuipeide@whut.edu.cn (P.C.); xuhaiqin@whut.edu.cn (H.X.); zouyingxue@whut.edu.cn (Y.Z.); yangxk@whut.edu.cn (X.Y.); 2School of Transportation, Southeast University, Jiulonghu, Nanjing 211189, China; 3Department of Civil Construction and Environmental Engineering, University of Alabama, Tuscaloosa, AL 35487, USA; samirkhanian@eng.ua.edu

**Keywords:** steel-slag powder, asphalt mortar, anti-aging properties, heavy-metal ions

## Abstract

The use of steel slag powder instead of filler to prepare asphalt mortar was beneficial to realize the effective utilization of steel slag and improve the performance of asphalt concrete. Nevertheless, the anti-aging properties of steel-slag powder–asphalt mortar need to be further enhanced. This study used antioxidants and UV absorbers in steel-slag powder–asphalt mortar to simultaneously improve its thermal-oxidation and UV-aging properties. The dosage of modifier was optimized by second-generation non-inferior sorting genetic algorithm. Fourier-Transform Infrared Spectroscopy, a dynamic shear rheometer and the heavy-metal-ion-leaching test were used to evaluate the characteristic functional groups, rheological properties and heavy-metal-toxicity characteristics of the steel-slag-powder-modified asphalt mortar, respectively. The results showed that there was a significant correlation between the amount of modifier and G*, δ, and the softening point. When the first peak appeared for G*, δ, and the softening point, the corresponding dosages of *x*_1_ were 2.15%, 1.0%, and 1.1%, respectively, while the corresponding dosage of *x*_2_ were 0.25%, 0.76%, and 0.38%, respectively. The optimal value of the modifier dosage *x*_1_ was 1.2% and *x*_2_ was 0.5% after weighing by the NSGA-II algorithm. The asphalt had a certain physical solid-sealing effect on the release of heavy-metal ions in the steel-slag powder. In addition, the asphalt structure was changed under the synergistic effect of oxygen and ultraviolet rays. Therefore, the risk of leaching heavy-metal ions was increased with the inferior asphalt-coating performance on the steel-slag powder.

## 1. Introduction

Asphalt pavement has been widely used in expressways due to its advantages of flat surface without joints, low noise, fast traffic recovery, and convenient maintenance [1,2]. By the end of 2020, the total highway mileages were 5,198,100 km in China, which is an increase of 185,600 km from the end of the previous year. However, due to complex traffic loads and climatic conditions, physicochemical reactions such as dehydrogenation, polycondensation, and oxidation [3] of asphalt pavement have been produced on account of the coupling of heat and oxygen [4,5]. Its road performance has rapidly attenuated and deteriorated with the occurrence of stresses, such as stripping, potholing, and cracking [6]. Therefore, improving the anti-aging performance of asphalt-pavement materials is crucial for reducing the stress on asphalt pavement, prolonging its service life, and ensuring traffic safety [7,8,9].

The annual production of steel slag was more than 100 million tons with a low comprehensive-utilization rate of only about 30% [10]. The accumulation of steel slag causes water pollution, heavy-metal release and other problems by occupying a lot of land [11,12,13]. Therefore, an enormous challenge has been posed to local ecological development and resource reuse. Steel slag has the properties of a rough surface texture, high intensity, wear resistance, and high alkalinity [14,15,16], which make it an innovative substitute for natural aggregate in asphalt mixtures. Previous research has focused on the use of steel slag as an aggregate in asphalt mixtures while neglecting the replacement of natural mineral powder with steel-slag powder (SSP) [17]. The use of SSP in asphalt concrete is beneficial to the improvement of its effective-utilization rate, and could enhance the performance of bituminous concrete by increasing the stiffness of asphalt mortar [18]. However, the steel-slag powder–asphalt mortar (SSP–AM) has poor anti-aging properties, especially under UV radiation. The porous structure of slag powder leads to multiple reflections of UV radiation, which exhibits easier entry than heat and oxygen and induces more serious aging [19,20].

Nowadays, some researchers have improved the aging resistance by adding nanoparticles or polymer modification to asphalt [21,22]. Hu, ZH et al. [23] investigated the preparation of composite antioxidants from expanded graphite (EG) loaded with CaCO_3_ nanoparticles and Mg(OH)_2_ (EG/CaCO_3_/MH) in order to reduce the thermo-oxidative aging of the binder. The results showed that EG/CaCO_3_/MH had a synergistic inhibitory effect on the thermo-oxidative aging of asphalt cement. Yang, J et al. [24] evaluated the UV-aging resistance of titanium dioxide/polystyrene reduced graphene oxide (TiO_2_/PS-rGO) on SBS modified asphalt. It was found that the viscosity aging index and ductility retention of the TiO_2_/PS-rGO/SBS-modified binder were decreased by 65.36 and increased by 8.79, respectively, which indicates that the UV-aging resistance of the binder was improved by TiO_2_/PS-rGO. Rajib, A et al. [25] explored the feasibility of biochar in the retardation of the aging of rubberized and base asphalt binder. The results indicated that biochar could delay the UV aging of asphalt by reducing the rheological and chemical aging-indicators of asphalt. However, only the anti-aging properties of the binder were the concern of most researchers, and the research directions were only thermal-oxygen aging or ultraviolet aging. There have been few reports on the coupling of UV- and thermo-oxidative-aging resistance of asphalt mortar.

For SSP–AM, it was of great significance to explore a new method of simultaneously improving the thermal-oxidative- and UV-aging properties. Based on this, the antioxidants 1098 and UV-5411 were selected for asphalt mortar at the same time to solve this problem. The 1098 antioxidant is a typical high-temperature antioxidant with excellent refractory characteristics. The temperature of the asphalt mixture reaches as high as 160 °C during the mixing process, so the antioxidant needs to have excellent high-temperature resistance. The UV absorbent UV-5411 is an ultraviolet absorber with a high-efficiency light-stabilization effect. The ultraviolet rays are converted into heat by ultraviolet absorption through chemical action, thereby preventing further thermal-oxygen and ultraviolet aging of the asphalt. Nevertheless, in the multi-objective optimization of the modifier dosage, the experimental analysis samples were obtained by the researchers through orthogonal experiments. The combination of sample points obtained by this method did not always fill the total design space of the variables [26]. It was prone to the blind accumulation of sample points, resulting in a waste of time and resources. Therefore, an effective experimental design method was adopted to comprehensively investigate the performance parameter of asphalt modified by the two additives.

The application of numerical simulation, an approximate mathematical model and engineering optimization to solve multi-parameter and multi-objective optimization algorithms has become a research hotspot [27,28]. Kollmann, J et al. [29] used the optimization method of a two-dimensional finite-element model to simulate the generation and propagation of cracks in the asphalt mixture. It was illustrated that damage occurred between each successive element, resulting in relatively unconstrained crack growth. Sivilevicius, H et al. [30] applied numerical simulations of the composition to select the optimum added quantity of new and old materials on the road surface. The results showed that the algorithm can be used in the preliminary stage of the RHMA hybrid structure design. The amount of RAP could not exceed 8.43% when the RHMA mixture AC-16 was applied to the surface layer. The Latin hypercube design (LHD) is a random, multidimensional, stratified sampling method [31]. The n-dimensional space is evenly divided into m intervals, and independent equal probability sampling is performed in each sub-interval [32]. The uniformity of random LHD was improved by the Optimal Latin hypercube design (Opt LHD), making the fit of factors and responses more uniform and random within the design-space region [33].

Steel-slag powder–asphalt mortar (SSP–AM) has poor anti-aging properties. The research on the coupling of UV and thermal-oxidative-aging resistance of SSP–AM is still unclear. In the research on modifier dosage, orthogonal experiments tend to blindly accumulate sample points, resulting in a waste of time and resources, and the multi-objective optimization exploration of modifiers in SSP–AM has not been fully explored. The performance of SSP–AM has not been thoroughly studied. The leaching behavior of heavy metals from SSP–AM is still not known. Based on the above background, the purpose of this research was as follows:The dosage parameters of antioxidants and UV absorbers were selected as the inputs, and the sample points were established based on the Opt LHD design method.The softening point, phase angle, and complex shear modulus of asphalt mortar after thermal-oxygen and ultraviolet aging were selected as the outputs to establish a Kriging model [34].The multi-objective optimization analysis of antioxidants and UV absorbers was performed by using NSGA-II [35] to determine the optimal dosage of each modifier. Hence, a chemically reactive composite-modified asphalt mortar with good comprehensive performance was prepared.The optimized model was verified in accordance with the experimental results. The chemical and phase compositions of SSP were detected by X-ray fluorescence (XRF) and X-ray diffraction (XRD). The characteristic functional groups, rheological properties and heavy-metal-toxicity characteristics of the composite-modified asphalt mortar were investigated by Fourier-Transform Infrared Spectroscopy (FTIR), a dynamic shear rheometer (DSR) and the heavy-metal-ion-leaching test (TCLP), respectively.

## 2. Materials

### 2.1. Steel-Slag Powder

An experimental study was conducted on SSP of Baotou Iron and Steel Co., Ltd., Baotou, Inner Mongolia, China. The jaw crusher was used to grind the steel slag to make its particle size less than 0.075 mm. Its basic properties are shown in Table 1, and it can be seen that the indexes met the requirements of the test methods of aggregate for highway engineering. Meanwhile, XRF was used to detect the chemical composition as shown in Table 2. The main components of the SSP were Al, Si, Fe, Ca, Mg and Mn respectively.

### 2.2. Asphalt

The AH-70 base asphalt was produced by Guochuang Co., Ltd., Wuhan, Hubei, China. The essential performance indexes of asphalt were tested in accordance with Standard Test Methods of Bitumen and Bituminous Mixtures for Highway Engineering (JTG E20-2011). The test results met the requirements as shown in Table 3.

### 2.3. Antioxidant

The German BASF 1098 antioxidant used in this article was a nitrogen-containing hindered phenolic antioxidant with low volatility, resistance to extraction, non-toxicity, heat-resistant oxidation, and other properties. The 1098 antioxidant in this paper was selected because it is a typical high temperature antioxidant with excellent refractory characteristics. The asphalt mixtures were mixed at 160 °C, so the antioxidant should have excellent high-temperature resistance. The photothermal aging resistance of polymer materials was effectively improved on account of its functions such as decomposing hydroperoxide, trapping free radicals, and trapping singlet oxygen. Its addition to asphalt was expected to trap thermal or photoinduction-free radicals based on the light-stabilization effect, thereby preventing further aging reactions of the asphalt. Its decomposition chemical equation and basic physical properties are presented in Figure 1 and Table 4, respectively.

### 2.4. Ultraviolet Absorbent

The BASF UV-5411 used in this study was a hydroxyphenyl benzotriazole ultraviolet absorber, which is an ultraviolet absorber with high-efficiency light stabilization. The ultraviolet rays were converted into heat by UV-5411 through chemical action, thereby preventing further thermal-oxygen and ultraviolet aging of the asphalt. Its decomposition chemical equation and basic physical properties are shown in Figure 2 and Table 5, respectively.

### 2.5. Preparation of Steel-Slag-Powder-Modified Asphalt Mortar

In this paper, asphalt mortar was prepared by using steel-slag powder instead of natural mineral powder. The density of mineral powder and steel-slag powder is quite different. Taking limestone mineral powder as an example, its apparent relative density is 2.7, while the apparent relative density of steel-slag powder is 3.2. Therefore, the volume ratio was used to mix the asphalt mortar, and the volume percentage of asphalt: steel-slag powder was 1:0.3. In addition, Wei et al. [36] determined that the amount of asphalt and fillers was determined by the qualified asphalt concrete mix proportion, while the asphalt aggregate ratio was 4.9%. Therefore, the volume ratio of fillers to asphalt was determined as 0.3.

The preparation process was shown in Figure 3 as follows: First, the steel-slag powder and asphalt were placed in an oven to be heated (AH-70 asphalt at 135 °C and steel-slag powder at 150 °C). Secondly, the asphalt was heated in an oil bath at 160 °C, and then the steel-slag powder (the volume ratio was 0.3) and modifiers (UV-5411 was 0–2.4 wt% and antioxidant 1098 was 0–4.8 wt%) were added with shearing at a low speed of 500 rpm for 15 min. Then, the machine was sheared at a high speed of 4000 rpm for 30 min to fully mix the steel-slag powder and modifiers with the asphalt. Finally, the prepared steel-slag-powder-modified asphalt mortar was transferred to a storage tank for use.

Since the density of steel-slag powder was much higher than that of 70# base asphalt, the steel-slag powder would gradually segregate to the lower part of the asphalt mortar during the storage process, which resulted in poor uniformity. Therefore, the principle of current production was adopted during the use of steel-slag-powder-modified asphalt mortar in order to reduce the error caused by the experiment.

## 3. Experimental Methods

### 3.1. Optimization Test of Dosage Simulation

#### 3.1.1. Experimental Design Based on Opt LHD Sampling

The basic principle of LHD sampling is to evenly divide each dimensional coordinate interval [*x*_k_^min^, *x*_k_^max^], i ∈ [1, n] into m intervals in the n-dimensional space, and each small interval is denoted as [*x*_k_^i−1^, *x*_k_^i^], i ∈ [1, m]. However, the uniformity of sampling points could not be considered in this sampling method, so the uniform space filling of sample points was not filled. Therefore, the Opt LHD sampling method was adopted to improve the uniformity of random LHD. The fit of factors and responses was more uniform and random within the design-space region, resulting in higher computational accuracy for the design of the experimental methods. In this paper, the dosage of antioxidant 1098 *x*_1_ ∈ (0, 4.8) and the UV absorber 5411 *x*_2_ ∈ (0, 2.4) were evenly divided into 25 sample spaces. The distribution of LHD and Opt LHD sampling designs are shown in Figure 4a,b, respectively. It can be seen that some regional sampling points were too concentrated in LHD sampling, while others were lost. However, the distribution of test points was more uniform in the Opt LHD.

#### 3.1.2. Experimental Design Based on Kriging Model

Kriging, also known as spatial local interpolation, is an unbiased optimal estimation method for regional variables in a finite area based on variogram theory and structural analysis. The *x*_1_ and *x*_2_ were set as the inputs; G*, δ, and the softening point were set as the outputs. A Kriging model was established for 25 sample data with its basic principle as follows:

*x*_0_ is set as the unobserved point that needs to be estimated, *x*_1_, *x*_2_, …, *x*_N_, are set as the observed points around it, and the observed values are correspondingly *y*(*x*_1_), *y*(*x*_2_), …, *y*(*x*_N_). The estimate of the unmeasured point is denoted as *ỹ*(*x*_0_), which is obtained by the weighted sum of the known observations of the adjacent observation points:(1)$$\tilde{y}(x_{0})=\sum_{i=1}^{N}\lambda_{i}y(x_{i})$$

Among them, *λ_i_* is the undetermined weighting coefficient. The key to the Kriging interpolation is to calculate the weight coefficient *λ_i_*, which must satisfy two conditions:

1The unbiased estimates: the true value of the evaluation point is set to *y*(*x*_0_). The $y\left(x_{i}\right)$, $\tilde{y}\left(x_{0}\right)$ and *y*(*x*_0_) can be regarded as a random variable.
(2)$$E\left[\tilde{y}(x_{0})-y(x_{0})\right]=0,\;\sum_{i=1}^{N}\lambda_{i}=1$$

2The variance of the difference between the true value *y*(*x*_0_) and the estimated value is minimal.
(3)$$D\left[\tilde{y}(x_{0})-y(x_{0})\right]=\min$$
(4)$$D\left[\tilde{y}(x_{0})-y(x_{0})\right]=-\sum_{i=1}^{N}\sum_{j=1}^{N}\lambda_{i}\lambda_{j}\gamma (x_{i},x_{j})+2\sum_{i=1}^{N}\lambda_{i}\gamma (x_{i},x_{0})$$
where, *γ*(*x_i_*, *x_j_*) represents the semivariance value of the parameter when the distance between the two points *x_i_* and *x_j_* is used as the distance h. The *γ*(*x_i_*, *x*_0_) is the semivariance value of the parameter when the distance between the two points *x_i_* and *x*_0_ is used as the distance h.

#### 3.1.3. The Multi-Objective Optimization Based on NSGA-II

The simultaneous optimization of multiple sub-objectives is a multi-objectives optimization problem. In most cases, it is generally impossible to concurrently achieve the optimum for multiple sub-goals. The ultimate goal of solving multi-objectives optimization problems is to harmonize various objectives, hence each sub-objective is as optimal as possible. In the non-dominated ranking, NSGA-II was selected to approach the individual of the Pareto front, which enhanced its ability to advance. In the Pareto optimal relationship, the individuals in the group were compared according to their target function vector and divided into multiple frontier layers that were controlled in sequence. NSGA-II is widely used due to its advantages of solving the Pareto solution set with good accuracy and dispersion. NSGA-II was used to optimize the Kriging approximation model in this research. The constraints, optimization objectives, lower and upper bounds were set according to the established mathematical model to obtain a set of solutions that all satisfy the conditions. The main process was in Figure 5 as follows:The initial population *P*_0_ was randomly generated and a new population *Q*_0_ was generated through selection, intersection and variation. The population *R*_0_ WAs obtained by merging *P*_0_ and *Q*_0_.*R*_t_ was sorted by non-inferiority to obtain non-inferior front segments *F*_1_, *F*_2_, ….*F*_i_ was sorted by crowding distance, and the better individuals and the previous segments *F*_1_, *F*_2_, …, *F*_i−1_ were selected to form N individuals and population *P*_t+1_.Population *P*_t+1_ was replicated, crossed and deformed to form population *Q*_t+1_. If the termination conditions were met, then it ended. Otherwise, it proceeded to step 2 to continue execution.

#### 3.1.4. Optimize Experimental Design

The sampling points were established based on Opt LHD, as shown in Table 6. The steel-slag-powder-modified asphalt mortar (SSP-MAM) with different contents of antioxidant and UV absorber was prepared to undergo TFOT short-term aging and seven-day UV aging. The complex shear modulus (G*), softening point and phase angle (δ) of the aged SSP-MAM were recorded by observation.

Based on the experimental basis and the literature of the laboratory, the content of antioxidant 1098 was in the range of 0–4.8%, represented by *x*_1_; the content of UV-5411 was in the range of 0–2.4%, represented by *x*_2_.

### 3.2. Properties of SSP

Empyrean XRD (Panaco, The Netherlands) and Axios XRF (Almelo, The Netherlands) were used to characterize the physicochemical properties of the SSP.

### 3.3. Aging-Test Method

#### 3.3.1. Short-Term Thermal-Oxygen-Aging Test

The 85-type asphalt film oven was used for the TFOT aging test to simulate the short-term thermal-oxygen-aging behavior of SSP-MAM in the mixing, paving, and compaction processes. The test temperature was controlled at 163 ± 1 °C, the rotating speed of the turntable was set at 5.5 ± 1 r/min, the tilt angle of the turntable and the horizontal plane was not more than 3°, and the aging time was 5 h.

#### 3.3.2. Accelerated-UV-Aging Test

The LHX-205 type intelligent numerical-control ultraviolet-aging test box was used to simulate accelerated-ultraviolet-aging experiments. The SSP-MAM sample after short-term thermal-oxygen aging was placed on the horizontal turntable of the ultraviolet-aging box. The ultraviolet radiation intensity on the surface of the asphalt sample was 30 W/cm^2^ by adjusting the height of the horizontal turntable. The wavelength of ultraviolet light was 365 nm, the experiment temperature was set to 60 °C, and the experiment time was seven days.

### 3.4. Asphalt Mortar Characteristic Analysis Method

#### 3.4.1. Functional Group Characteristics of Asphalt Mortar

In the FTIR, the molecular structure was observed by measuring the vibrational and rotational spectra of molecules. A Nexus intelligent FTIR was adopted to analysis the functional groups of SSP-MAM. First, the asphalt/carbon disulfide solution with a mass concentration of 5% was configured. Then the asphalt sample was dissolved in the carbon-disulfide solution, and the prepared solution was dropped onto the KBr wafer. The SSP-MAM sample film could be obtained after the complete volatilization of the carbon-disulfide solution. The scanning beam range of the infrared spectrometer was 4000–400 cm^−1^, and the number of scans was 64.

#### 3.4.2. Rheological Properties of Asphalt Mortar

The dynamic shear rheometer (DSR) is an instrument applied to analyze the rheological properties of polymer materials. It has been widely used due to its good experimental accuracy and repeatability, since SHRP recommended using it to test the rheological properties of asphalt in 1993. The rheological properties of SSP-MAM samples were tested by MCR-102 DSR (Anton Paar, Graz, Austria). The rheological parameters at different temperatures were scanned under the strain-control mode. The scanning frequency was controlled as 10 rad/s, the temperature scanning range was 30–80 °C, the upper and lower plate diameter was 25 mm, the heating rate was 2 °C/min, and the plate spacing (asphalt film thickness) was 1 mm.

### 3.5. Leaching Behavior of Asphalt Mortar

TCLP developed by the US Environmental Protection Agency (EPA) was adopted for the toxicity-extraction experiment of steel slag in this research. The leaching agent was used to adjust the pH of the solid waste for tumble-extraction experiments in order to implement the Resource Conservation and Regeneration Act (RCRA) for the management of hazardous and solid waste. Simultaneously, the leaching content of inorganic components in solid waste was simulated by judging its contamination capacity.

In this experiment, the leaching behavior of heavy-metal ions from SSP and SSP-MAM was investigated. SSP-MAM was specially prepared into particles smaller than 9.5 mm to meet the requirements of the test samples in TCLP. The specific experimental steps in Figure 6 were as follows: First, 50.0 g of the particle was weighed and placed in a 2 L polyethylene bottle for later use. Secondly, the acetate buffer solution with pH value of 2.88 ± 0.05 was prepared, and the corresponding amount of acetate buffer solution was added according to the liquid- solid ratio of 20:1 (L/kg). Then, the polyethylene bottle was placed on the inversion shaker, the rotation speed of the shaker was set to 30 rpm, the shaking time was 18 ± 2 h, and the temperature was 23 ± 2 °C. Finally, the mixture was allowed to stand after shaking, and the clarified filtrate was collected in a centrifuge tube with filter paper for next step of testing. The leaching concentration of heavy-metal elements was determined by atomic absorption spectrometry.

### 3.6. Experiment Plan

The experimental plan was shown in Figure 7.

## 4. Results and Discussions

### 4.1. Optimization Test of Dosage Simulation

The partial-approximation model, contour plot, and R-Squared accuracy of G* and *x*_1_, *x*_2_ are presented in Figure 8a–c, and so on for phase angle and softening point. From Figure 8c,f,i, the values of R^2^ can be obtained as 0.96652, 0.95828, and 0.95732, respectively, where the blue line represents the actual average value, indicating that the fitting accuracy and reliability were high. The fit of anisotropy and exponential functions were selected as the Kriging model by comparing the output R^2^ values. Figure 8b,e,h are contour maps of the projections of Figure 8a,d,g on the horizontal plane, respectively. The denser the contour lines and the greater the slope of the fitted surface, the more significant the influence of this factor was. The contour lines in the figure were dense and steep, indicating that there was significant correlation between the amount of modifier and G*, δ, and the softening point. When the first peak appeared, for G*, δ, and the softening point, the corresponding dosages of *x*_1_ were 2.15%, 1.0%, and 1.1%, respectively. It showed that the effect of antioxidants on δ was more significant, that is, less antioxidant could change δ. Meanwhile, the corresponding dosages of *x*_2_ were 0.25%, 0.76%, and 0.38%, respectively. It was clarified that the effect of the UV absorber on G* was more significant, that is, less UV absorber could change G*.

In this paper, the softening point and phase angle were selected as research objects as examples. The optimized Pareto front is shown in Figure 9. The parameters of the NSGA-II algorithm were set as: population size was 100, evolutionary generation was 100, hybridization-distribution coefficient was 20, hybridization probability was 0.9, mutation-distribution coefficient was 100, and mutation probability was 0.9. All the points in Figure 9 were qualified non-inferior solutions. It was necessary to select the appropriate solution in order to determine the final optimization scheme. The individuals with larger crowding distances had greater advantages in accordance with the optimization strategy of the NSGA-II algorithm. The optimal value of the modifier dosage *x*_1_ was 1.2%, and *x*_2_ was 0.5% after weighing by the NSGA-II algorithm.

### 4.2. Properties of SSP

The mineral phase of SSP was characterized by XRD as shown in Figure 10. It can be seen from the diffraction pattern that there were a variety of diffraction peaks with serious overlap, which indicates that the types of minerals contained in the steel slag were quite complex. The strongest diffraction peaks and three strong peaks appearing at 2θ of 32.961 and 33.067 belonged to silicate minerals, indicating that C_3_S and C_2_S were the main mineral phases of steel slag.

### 4.3. Functional-Group Characteristics of Asphalt Mortar

Infrared spectroscopy tests were carried out on modified asphalt mortar (MAM) and unmodified asphalt mortar (UAM) to investigate the change characteristics of functional groups during thermal-oxygen and ultraviolet aging. The characteristic peaks of oxygen-containing functional groups were more distinct due to the absorption of oxygen during aging. The formula to calculate the index of the characteristic functional group carbonyl (C=O) and sulfoxide (S=O) of asphalt were as follows:(5)$$I_{C=O}=\frac{S_{{1700\;\mathrm{cm}}^{-1}}}{S_{{2000\sim 600\;\mathrm{cm}}^{-1}}}\times 100\%$$
(6)$$I_{S=O}=\frac{S_{{1030\;\mathrm{cm}}^{-1}}}{S_{{2000\sim 600\;\mathrm{cm}}^{-1}}}\times 100\%$$
where $S_{{1700\;\mathrm{cm}}^{-1}}$ is the area of the carbonyl peak band centered around 1700 cm^−1^, $S_{{1030\;\mathrm{cm}}^{-1}}$ is the area of the sulfoxide peak band centered around 1030 cm^−1^, and $S_{{2000\sim 600\;\mathrm{cm}}^{-1}}$ is the area of the spectral bands between 2000 and 600 cm^−1^.

In the UAM, the value of *I*_S=O_ and *I*_C=O_ were 1.56% and 1.3%, respectively, while the values of *I*_S=O_ and *I*_C=O_ in the MAM were 0.9% and 1.1%, respectively. It revealed that the content of oxygen-containing functional groups and the degree of photo-oxidation of the MAM were declined. Therefore, the thermal-oxygen- and ultraviolet-aging resistance of the MAM was improved.

In the Figure 11, the characteristic absorption peaks of *I*_S=O_ and *I*_C=O_ in UAM appeared at 1033 cm^−1^ and 1692 cm^−1^, respectively, while in MAM they appeared at 1676 cm^−1^ and 998 cm^−1^, respectively. The reason for the different degrees of bathochromic shift of the two characteristic absorption peaks was that the antioxidants and UV absorbers contained auxiliary color groups such as non-bonded-electron heteroatom-saturated groups. Simultaneously, the absorption peaks being shifted to long wavelengths under the combined action of conjugation and auxochrome groups was attributed to the presence of aromatic heterocycles in antioxidants and UV absorbers.

In addition, the characteristic peaks of C-H stretching vibration of N-H and aromatic hydrocarbon benzene ring in MAM were present at 3644 cm^−1^ and 680 cm^−1^, respectively, but they did not appear in UAM. This illustrated that antioxidants and UV absorbers had chemical reactions with asphalt to generate new functional groups during the aging process of asphalt, rather than simple physical blending [37].

### 4.4. Rheological Properties of Asphalt Mortar

Figure 12 showed that G* decreased with the increase in temperature, indicating that the rheological properties of SSP-MAM had obvious temperature dependence. The transition of asphalt from a highly elastic state at low temperature to a viscous fluid state at high temperature due to the increase in the free volume of asphalt with increasing temperature. Therefore, the decrease in the maximum shear stress and the increase in the maximum shear strain of the asphalt led to the decrease in G*. Meanwhile, the G* of the UAM was larger, which was due to the oxidation reaction of the asphalt film in contact with the oxygen in the air at high temperature. The content of aromatics in asphalt was decreased, while pectin and asphaltene was increased. Therefore, a molecular weight migration from small to large occurred between the asphalt components. The asphalt was hardened by the reduction in plastic flow deformability based on the increase in the overall molecular weight of the asphalt. The G* in MAM was smaller on account of the modifier chemically reacting with oxygen, thereby preventing the contact of the bitumen film with oxygen.

The anti-aging ability of bitumen was enhanced with the increase in phase-angle ratio and the decrease in viscosity loss during the aging process. It can be observed that the phase angle increased with the increase in temperature, which was larger in MAM than in UAM. It illustrated that the free radicals generated by the asphalt monomer due to thermal oxygen and light were captured by antioxidants, while the ultraviolet rays were converted into heat by ultraviolet absorbent through chemical action, thus preventing further thermal-oxygen and ultraviolet aging of the asphalt.

### 4.5. Leaching Behavior of Asphalt Mortar

Figure 13 presents the leaching results of heavy-metal ions. It can be clearly discerned that the leaching concentrations of As, Cr^6+^ and Cu were higher, followed by Zn, Pb, Ni, and finally Cd. Meanwhile, the leaching concentration of heavy-metal ions in the SSP was the largest, followed by the aged SSP-MAM, and finally the unaged SSP-MAM.

A possible cause was that the SSP was coated with chemically stable bitumen to prevent it from reacting with the acidic extractant. On the other hand, the high-polarity water could not fully contact the SSP on account of the low polarity of the asphalt. Therefore, asphalt had a certain physical solid-sealing effect on the release of heavy-metal ions in the SSP, and further reduced the leaching heavy-metal-ion content by reducing the chance of contact between the SSP and the extraction solution. In addition, the encapsulation ability of the asphalt to the heavy-metal ions in the SSP was also imparity. The leaching amount of As and Pb decreased the most, which were 47.8% and 46.4%, respectively, followed by Cr^6+^ (22.6%) and Cd (21.2%), Zn (10.6%), and finally, Ni (7.4%) and Cu (5.2%).

Obviously, the aged SSP-MAM was larger than the unaged SSP-MAM in terms of ionic leaching concentration. The reason was that the asphalt structure was changed under the synergistic effect of oxygen and ultraviolet rays. The flexibility and ductility of the asphalt became depraved with the decrease in light components and the increase in asphaltene. At this point, the risk of leaching heavy-metal ions increased with the inferior asphalt coating performance on SSP.

## 5. Conclusions

SSP was used to replace natural mineral to prepare asphalt mortar in this research, and the thermal-oxidation- and ultraviolet-aging resistance of SSP-MAM was evaluated. The multi-objective optimization analysis of antioxidants and UV absorbers was carried out using the NSGA-II. A chemically reactive composite-modified asphalt mortar with good comprehensive properties was prepared. The optimization model was verified according to the experimental results. Based on the above research content, the main conclusions of this paper were as follows:The denser the contour lines and the greater the slope of the fitted surface, the more significant the influence of this factor was. The contour lines were dense and steep, indicating that there was significant correlation between the dosage of modifier and G*, δ, and the softening point. When the first peak appeared, for G*, δ, and the softening point, the corresponding dosages of *x*_1_ were 2.15%, 1.0%, and 1.1%, respectively, while the corresponding dosages of *x*_2_ were 0.25%, 0.76%, and 0.38%, respectively. All the points in the Pareto front were qualified non-inferior solutions. The individuals with larger crowding distances had greater advantages in accordance with the optimization strategy of the NSGA-II algorithm. The optimal value of the modifier dosage *x*_1_ was 1.2%, and *x*_2_ was 0.5% after weighing by the NSGA-II algorithm.In the UAM, the value of *I*_S=O_ and *I*_C=O_ were 1.56% and 1.3%, respectively, while the values of *I*_S=O_ and *I*_C=O_ in the MAM were 0.9% and 1.1%, respectively. This revealed that the content of oxygen-containing functional groups and the degree of photo-oxidation of the MAM were declined. The characteristic absorption peaks of sulfoxide and carbonyl group of the MAM had different degrees of bathochromic shift.The complex shear modulus was smaller and the phase angle was larger in MAM. The contact of the bituminous membrane with oxygen was prevented as the modifier reacted chemically with oxygen. This illustrated that the free radicals generated by the asphalt monomer due to thermal oxygen and light were captured by antioxidants, while the ultraviolet rays were converted into heat by ultraviolet absorbent through chemical action, thus preventing further thermal-oxygen and ultraviolet aging of the asphalt.The leaching concentration of heavy-metal ions in the SSP was the largest, followed by the aged SSP-MAM, and finally the unaged SSP-MAM. The asphalt had a certain physical solid-sealing effect on the release of heavy-metal ions in SSP. In addition, the encapsulation ability of asphalt for heavy-metal ions in SSP was also insufficient. The leaching amount of As and Pb decreased the most, which were 47.8% and 46.4%, respectively. The asphalt structure was changed under the synergistic effect of oxygen and ultraviolet rays. Therefore, the risk of leaching heavy-metal ions was increased with the inferior asphalt coating performance on SSP.

The UV-5411 and antioxidant 1098 increased the thermal oxygen and UV-aging resistance of SSP-MAM. Therefore, the mixing performance of the asphalt mixture was improved and its service life was prolonged. The solid-sealing effect of asphalt on SSP reduced the risk of leaching of heavy-metal ions, thus improving the application potential of SSP instead of mineral powder. However, due to the high economic cost of UV absorbers and antioxidants, their large-scale applications are limited. Therefore, future research can begin with reducing the dosage of modifiers, finding cheap replacement materials with excellent performance, or introducing new materials to reduce the dosage of UV absorbers and antioxidants.

## Figures and Tables

**Figure 1 materials-15-03635-f001:**
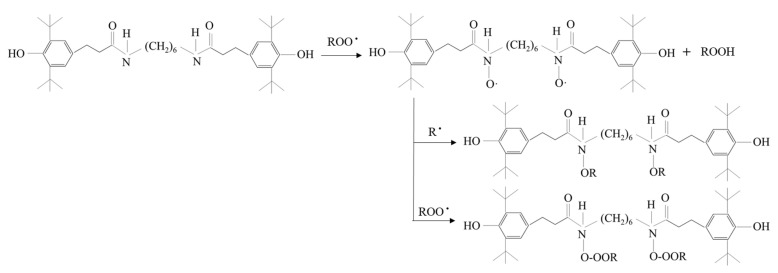
The decomposition chemical equation of antioxidant 1098.

**Figure 2 materials-15-03635-f002:**
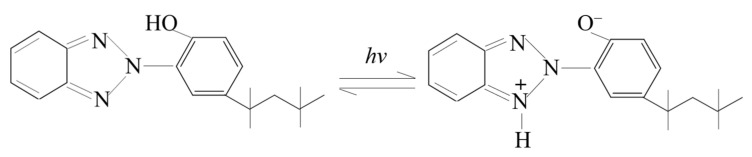
The decomposition chemical equation of UV-5411.

**Figure 3 materials-15-03635-f003:**
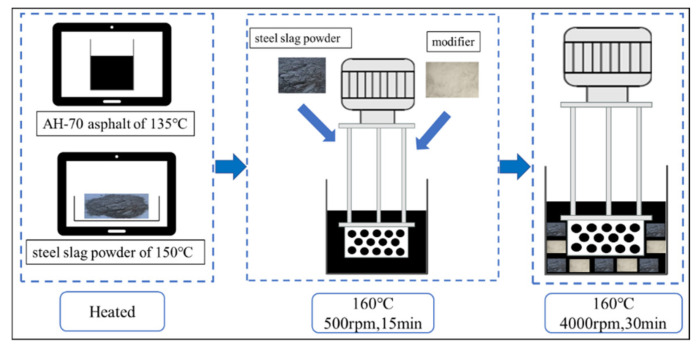
The preparation of steel-slag-powder-modified asphalt mortar.

**Figure 4 materials-15-03635-f004:**
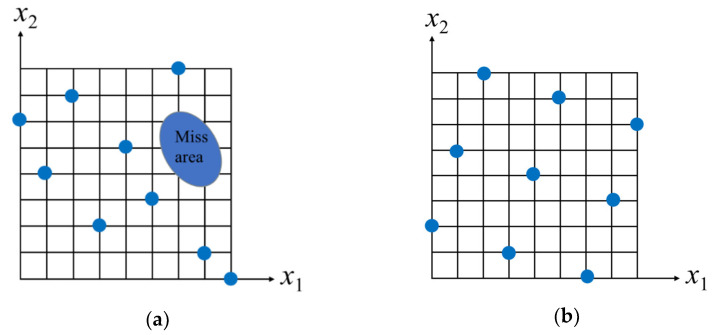
The distribution of LHD and Opt LHD sampling designs. (**a**) Random LHD sampling, (**b**) Opt LHD sampling.

**Figure 5 materials-15-03635-f005:**
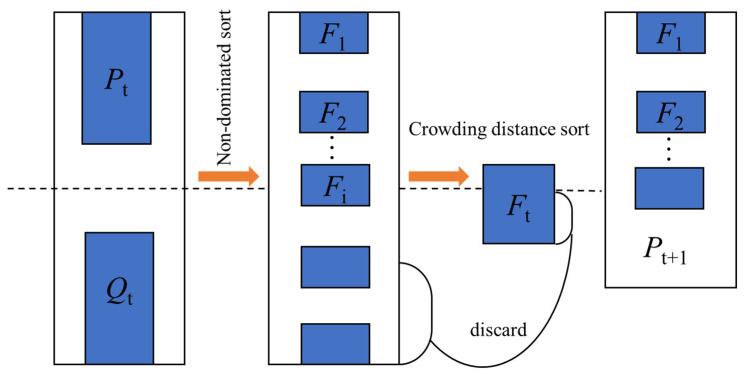
Flowchart for solving the Pareto solution set.

**Figure 6 materials-15-03635-f006:**
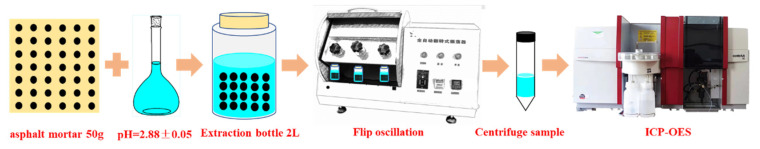
Procedures of TCLP.

**Figure 7 materials-15-03635-f007:**
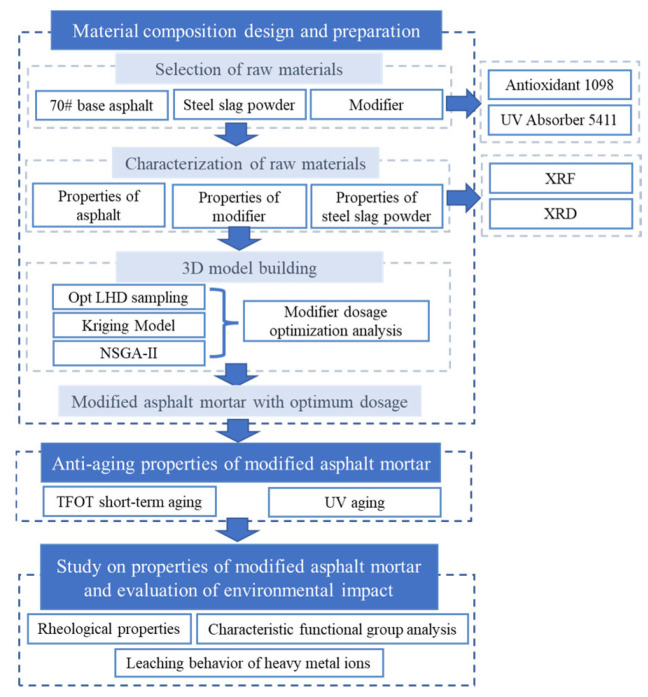
Experiment plan.

**Figure 8 materials-15-03635-f008:**
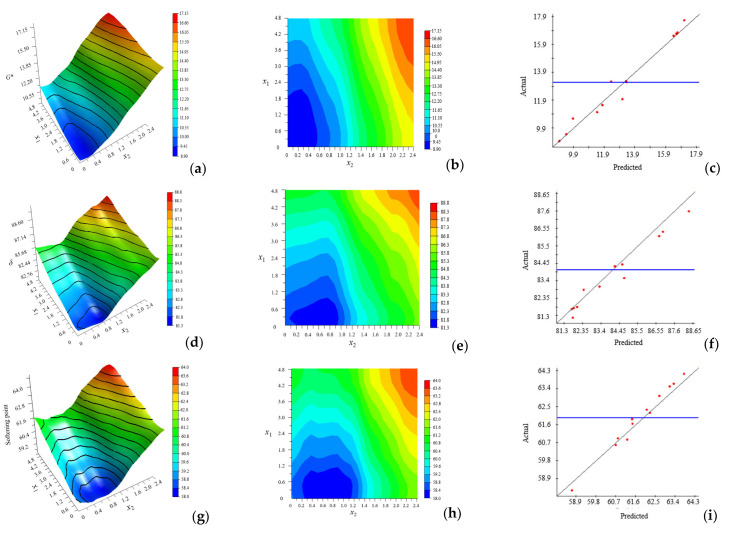
Partial-approximation model, contour plot, and R-squared accuracy of output and input. (**a**) Partial-approximation model of G*, (**b**) Contour plot of G*, (**c**) R-Squared accuracy of G*, (**d**) Partial-approximation model of δ, (**e**) Contour plot of δ, (**f**) R-Squared accuracy of δ, (**g**) Partial-approximation model of softening point, (**h**) Contour plot of softening point, (**i**) R-Squared accuracy of softening point.

**Figure 9 materials-15-03635-f009:**
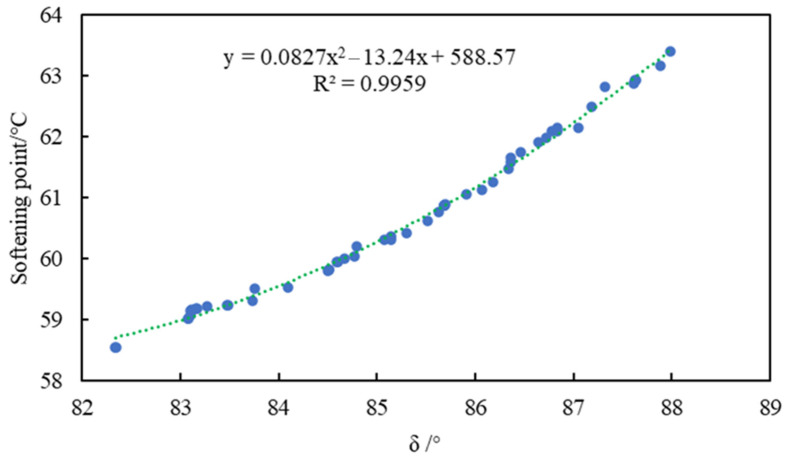
Pareto front.

**Figure 10 materials-15-03635-f010:**
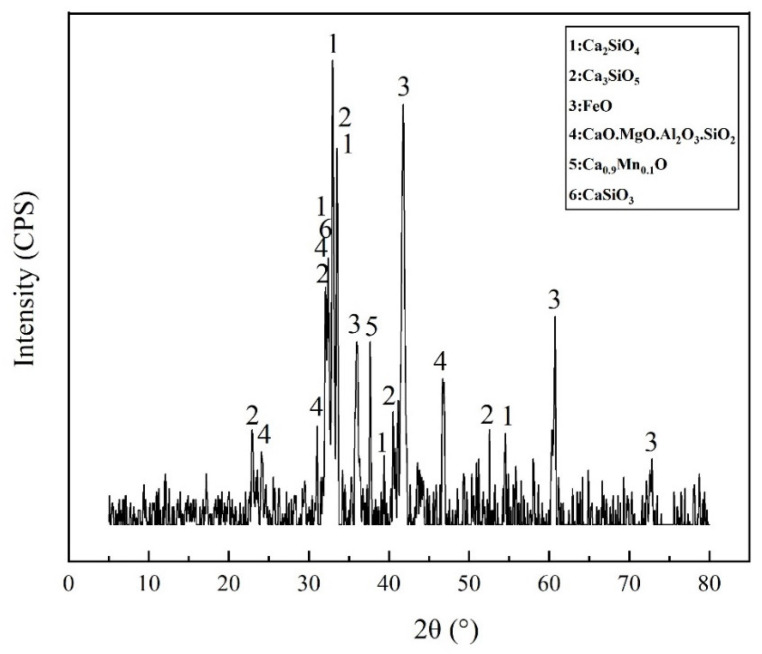
The XRD of SSP.

**Figure 11 materials-15-03635-f011:**
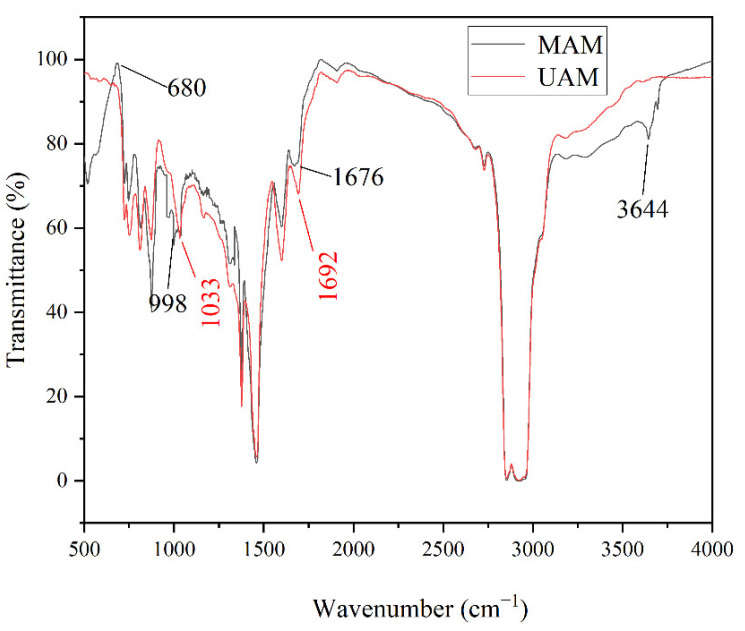
Infrared analysis of asphalt mortar.

**Figure 12 materials-15-03635-f012:**
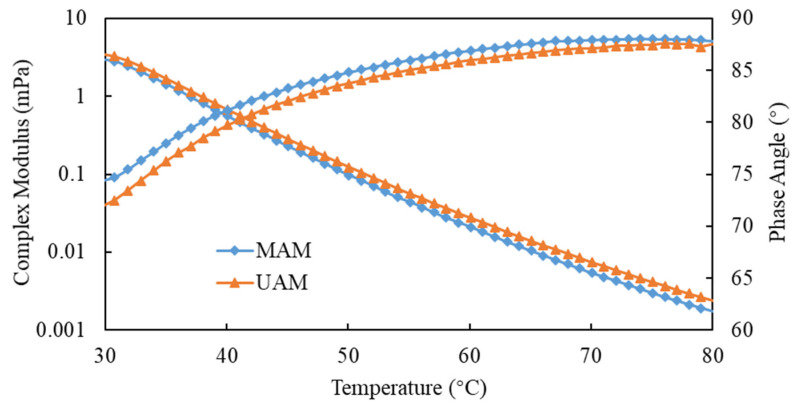
Rheological properties of asphalt mortar.

**Figure 13 materials-15-03635-f013:**
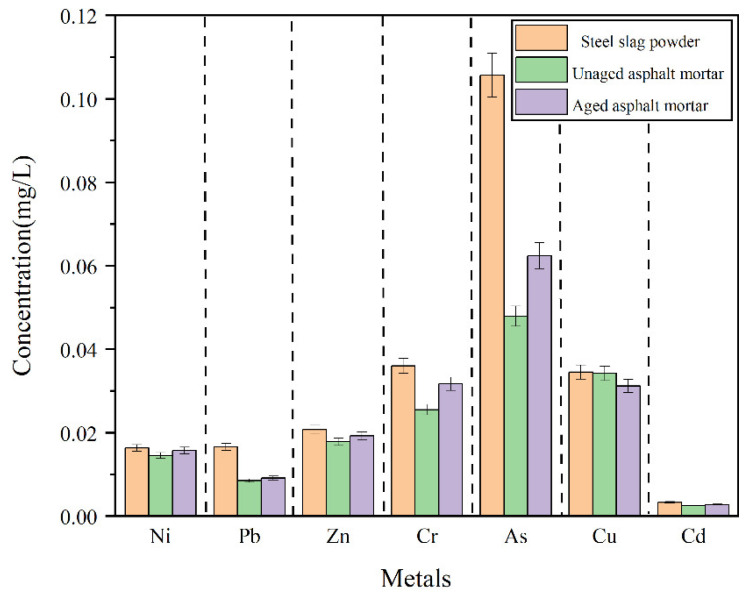
Heavy-metal leaching from asphalt mortar.

**Table 1 materials-15-03635-t001:** Basic properties of SSP.

Properties	Tested Value	Specifications
Density (g/cm^3^)	3.57	≥2.9
Los Angeles abrasion (%)	8.5	≤28
Specific surface area (m^2^/g)	1.95	-
Water absorption ratio (%)	0.68	≤1.0
Crush values	12.7	≤26

**Table 2 materials-15-03635-t002:** Chemical composition of SSP.

Compounds	CaO	Fe_2_O_3_	SiO_2_	MgO	Al_2_O_3_	MnO	TiO_2_	P_2_O_5_	Others
Steel slag	35.5	33.3	14.0	4.9	2.4	4.5	1.1	2.2	2.1

**Table 3 materials-15-03635-t003:** Physical information of asphalt.

Information	Unit	Result	Requirements
Penetration (25 °C, 5 s, 100 g)	0.1 mm	67.0	60.0–80.0
Softening point	°C	48.3	≥45.0
Ductility (5 °C, 5 cm/min)	cm	156.7	≥100.0
Density (15 °C)	g/cm^3^	1.035	-
Solubility (Trichloroethylene)	%	99.8	≥99.0

**Table 4 materials-15-03635-t004:** Basic physical properties of antioxidant 1098.

Information	Unit	Result
Melting point	°C	156–161
Flashing point	°C	282
Density (20 °C)	g/cm^3^	1.045
Solubility (Water)	%	0.01

**Table 5 materials-15-03635-t005:** Basic physical properties of UV-5411.

Information	Unit	Result
Melting point	°C	103.0–105.3
Flashing point	°C	>150
Density (20 °C)	g/cm^3^	1.18
Solubility (Water)	%	<0.01

**Table 6 materials-15-03635-t006:** Opt LHD experimental design and results.

Number	Modifier Dosage		Measured Value		
	*x*_1_ (%)	*x*_2_ (%)	G* (64 °C)/kPa	δ (°)	Softening Point (°C)
1	0	1.2	9.099	87.3	58.3
2	0.6	1.5	11.324	85.8	59.9
3	3.2	0.6	10.073	86.8	59.0
4	0.4	0.9	8.983	88.6	58.2
5	1.2	0	10.434	85.5	59.3
6	2.4	2.4	13.368	83.3	61.5
7	0.2	1.3	10.454	84.6	59.3
8	2.2	1.8	11.437	84.0	60.0
9	0.8	1.1	11.675	84.1	60.2
10	3.8	0.3	11.371	85.9	60.0
11	3.6	2.2	13.341	83.0	61.4
12	4.6	1.0	13.314	83.4	61.4
13	1.8	0.7	12.064	83.4	60.5
14	2.6	2.1	13.721	82.7	61.7
15	4.2	0.5	12.475	83.9	60.8
16	4.8	2.3	15.631	82.5	63.1
17	2.8	0.8	13.388	83.2	61.5
18	1.0	0.2	11.946	83.6	60.4
19	1.6	0.4	13.511	82.9	61.6
20	4.4	1.4	15.901	82.2	63.3
21	3.0	1.6	16.760	81.8	64.0
22	3.4	2.0	16.083	81.7	63.5
23	2.0	0.1	12.395	83.3	60.7
24	4.0	1.7	16.020	81.8	63.4
25	1.4	1.9	17.193	81.3	64.3

## Data Availability

The data presented in this study were available on request from the corresponding author.

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
