# Peer review of "Optimization of Asphalt-Mortar-Aging-Resistance-Modifier Dosage Based on Second-Generation Non-Inferior Sorting Genetic Algorithm"

_materials, 2022, doi:10.3390/ma15103635_

Round 1

Reviewer 1 Report

  1. Do all the changes highlighted in the attached paper.
  2. Please add the plates to show the preparation of materials, mixing process, casting and testing.

Reviewer 2 Report

This paper considers the Optimization of asphalt mortar aging resistance modifier dosage based on a second-generation non-inferior sorting genetic algorithm. The submitted article is interesting, original and within the scope of the journal but some changes should be addressed:

  1. The author needs to use an English language editorial service to check English language problems.
  2. The Authors should clearly present the preparation of the mixture with some photos or figures. In my opinion, you must put some figures inside the frame so that they will be good.
  3. I recommend using the same font and size for the axis titles of figures.
  4. The section on discussions is not written properly. Please revise it accordingly.
  5. Please rewrite the references according to journal instructions.

Reviewer 3 Report

  • Technical suggestion. The article should be entirely in accordance with the instructions (https://www.mdpi.com/journal/materials/instructions).
  • In the Introduction section, the last paragraph should contain the scientific contribution and scientific hypotheses of your research. Complete, further elaborate the scientific contribution and scientific hypotheses of your research. Be explicit. In addition to the goal of the research (which was written), the novelty in the context of the scientific contribution should be pointed out. Scientific contributions should be written based on the shortcomings of previous research in the literature. In this way, the authors will better emphasize novelty and scientific soundness.
  • Errors are an inevitable occurrence of all research. Errors can be random and systemic. Do they occur in your experiments. Analyze and discuss potential errors. Estimate the measurement uncertainty of the results. How much errors and uncertainties affect your results, applicability and the like.
  • The "2. Materials and Methods" section should not only list materials and methods. Every choice should be explained. Complete this section in detail.
  • Analyze and discuss possibilities of practical application.
  • Complete the conclusions with the limitations of the proposed methodology. Also write future research.
  • Generally, the quality of the writing could be improved.

Round 2

Reviewer 3 Report

The presented data are original and interesting. The manuscript has been significantly improved and is suitable for publication in the present Journal.